# Efficacy of acupuncture in the treatment of post-stroke depression: A study protocol of a randomized controlled trial

**Menghan Li**[1,2‡]*, **Dawei Ran**[1,2‡], **Xinming Yang**[1,2], **Yu Wang**[1,2], **Qian Zhu**[1,2], **Xiaoli Song**[1,2], **Lei Shi**[1,2]*, **Yuzheng Du**[1,2]

**1** Clinical Department of Acupuncture and Moxibustion, First Teaching Hospital of Tianjin University of Traditional Chinese Medicine, Tianjin, China, **2** National Clinical Research Center for Chinese Medicine Acupuncture and Moxibustion, Tianjin, China

‡ ML and DR contributed equally to this work and share first authorship
* limenghan01@126.com (ML); shilei1978ster@gmail.com (LS)

**Data Availability Statement:** No datasets were generated or analysed during the current study. All relevant data from this study will be made available upon study completion.

## Introduction

Post-stroke depression (PSD) is an emotional disease characterized by loss of interest and depression after a stroke. Acupuncture, one of the most critical non-drug therapies for the treatment of PSD, has significant clinical efficacy, but the mechanism is not fully understood. Previous study has shown that acupuncture can reduce the level of proinflammatory cytokines and increase the level of anti-inflammatory cytokines, suggesting that regulating the dynamic balance of inflammatory cytokines may be the basis for acupuncture to improve PSD symptoms.

## Methods and analysis

A total of 84 patients with PSD will be recruited and randomly assigned to one of two groups at a 1:1 ratio. Based on the conventional stroke treatment, the control group will receive XingnaoKaiqiao (XNKQ) acupuncture, and the experimental group will receive antidepressant acupuncture at the same time as XNKQ acupuncture. The intervention will last four weeks, and data will be collected before and after treatment. The Hamilton depression scale (HAMD-17) is the primary outcome measure, and the secondary outcome measures include the Self-rating Depression Scale (SDS), the National Institute of Health stroke scale (NIHSS), and the Modified Barthel Index (MBI). Serum IL-1β, IL-4, 5-HT, and BDNF will be used as laboratory indicators. The scales will be assessed at baseline, two weeks, and four weeks, and serum items will be measured at baseline and four weeks after treatment. This study will observe the clinical effect of acupuncture on PSD and the changes in serum-related inflammatory cytokines and explore the possible mechanism of acupuncture against depression from the perspective of inflammatory response.

**Funding:** This work was supported by the Tianjin Municipal Diversified Investment Fund for Applied Basic Research, Tianjin Municipal Bureau of Science and Technology (grant nos. 21JCQNJC01160). And Tianjin Education Commission Research Program Project (2021KJ150).

**Competing interests:** The authors have declared that no competing interests exist.

**Abbreviations:** PSD, post-stroke depression; XNKQ, xingnaokaiqiao (an acupuncture treatment); DSM- IV/V, Diagnostic and Statistical Manual of Mental Disorders; HAMD-17, Hamilton Depression Rating Scale; SDS, Self-rating depression scale; NIHSS, The National Institute of Health stroke scale; MBI, Modified Barthel Index; TCM, Traditional Chinese Medicine; CRF, case report forms; RCT, randomized controlled trial; EA, electro-acupuncture.

## Ethics and dissemination

This study protocol was approved by the Medical Ethics Committee of the First Teaching Hospital of Tianjin University of Traditional Chinese Medicine (item number: TYLL2023[Z] 004). Findings of the trial will be disseminated through peer-reviewed journals and scientific conferences.

## 1 Introduction

Post-stroke depression (PSD) is an affective disorder complication characterized by depression and loss of interest after a stroke, which has recently attracted more attention. According to statistics, the incidence rate of PSD in the first year of stroke is about 10% to 15%, and the incidence rate in 5 years is 39% to 52% [1]. In China, the incidence of PSD is around 34.9%, with women being more affected than men [2]. A recent animal experiment has emphasized the importance of sex specificity in determining PSD [3].

PSD manifests mainly as emotional apathy, weight change, sleep disturbance, fatigue, a sense of worthlessness, and anhedonia [4]. PSD is a negative prognosis in stroke recovery, frequently resulting in higher mortality, poorer functional recovery, significant cognitive impairment, and decreased quality of life [5].

It is essential to clarify the etiology and pathogenesis of PSD for its treatment. The occurrence of PSD is the result of a combination of social psychological factors and biological factors, including axis dysregulation, increased inflammatory factors, decreased monoamine levels, glutamate-mediated excitotoxicity, and abnormal neurotrophic responses [6, 7]. Among them, the theory of inflammatory response is one of the hotspots in the current research on the mechanism of PSD. Various inflammatory factors, primarily pro-inflammatory and anti-inflammatory cytokines that oppose each other, are implicated in the incidence of PSD, with different expressions in different phases of PSD. As antidepressant drugs can cause changes in inflammatory factors, changing the balance of inflammatory factors may be one of the strategies for treating PSD [8].

Currently, the first-line treatment for PSD is selective serotonin reuptake inhibitors (SSRIs), which can prevent and treat depression in PSD patients. However, adverse reactions such as cerebral hemorrhage, epilepsy, and gastrointestinal symptoms have increased to some extent [9]. Given the limitations of pharmacological treatment, several non-drug treatments, such as non-invasive brain stimulation, psychotherapy, exercise therapy, acupuncture, music, literature, and art, have been employed in the clinical treatment of PSD [10].

Acupuncture is an essential element of TCM (Traditional Chinese Medicine), and its therapeutic efficacy in the treatment of PSD is well-known. Acupuncture has been shown in several clinical studies to significantly improve the depressive symptoms of PSD patients, enhancing their quality of life [11–15]. Several systematic reviews have shown that acupuncture significantly improves HAMD and NIHSS scores [16, 17]. Another study found that electroacupuncture improves symptoms of PSD patients as does antidepressant drugs, with a higher safety [18]. Studies have found that acupuncture treatment of PSD may be an overall regulation of multiple aspects and targets. The specific mechanism basically involves neurotransmitter regulation, neuroendocrine disorders improvement, inflammatory factor reduction, oxidative stress reduction, neuron protection, and regeneration promotion [19]. Our previous studies have shown that acupuncture can reduce the levels of pro-inflammatory factors and increase the anti-inflammatory factors to regulate the dynamic balance of inflammatory factors, which

may be one of the mechanisms for acupuncture to improve PSD [20]. According to the preliminary clinical observation, acupuncture can improve the depressive symptoms of PSD patients, the motor function of limbs, and the quality of life [15].

At the moment, the mechanism of acupuncture treatment for PSD remains unknown. Although the research evidence from animals and humans has gradually increased in recent years, more is required to be firmly proven [21]. Most clinical trials are small in scale, so there is insufficient evidence for the true efficacy of PSD patients. Therefore, while evaluating the clinical effectiveness of acupuncture for PSD through this project, it is of great significance to observe the effect of acupuncture on the inflammatory factors of PSD patients, further explore the mechanism of acupuncture for PSD, and provide a theoretical basis for clinical acupuncture treatment of PSD.

## 2 Method and analysis

### 2.1 Study design

A single-center, prospective, parallel-group, randomized controlled trial will be conducted in the First Teaching Hospital of Tianjin University of Traditional Chinese Medicine. A total of 84 PSD patients will be recruited, with those who satisfy the inclusion criteria being randomly assigned to one of two groups. One group will receive XNKQ acupuncture mainly, and escitalopram will be taken orally, and the other group will receive antidepressant acupuncture in addition to the above acupuncture treatment. All acupuncture treatments will be given six times a week for four weeks (orally take escitalopram tablets daily for 4 weeks). Outcomes include scale indicators, serum laboratory indicators, and metabolomics analysis results. The primary outcome is depressive symptom severity and corresponding changes, as assessed by the Hamilton Anxiety Scale (HAMD-17). The secondary outcomes are the severity of neurological deficit symptoms, quality of life, laboratory indicators, and serum metabolomics analysis in PSD patients. These results will be assessed by the National Institutes of Health Stroke Scale (NIHSS) and Modified Barthel Index Scale (MBI), self-rating depression scale (SDS) for self-assessment, and laboratory indicators including the determination of serum IL-1β, IL- 10 5-HT, and BDNF levels. Evaluators will conduct assessments at baseline, week 2, and week 4 (serum indicators and metabolomics analysis will be assessed at baseline and week 4). The flow chart of this trial is displayed in **Fig 1**, and the enrollment schedule is illustrated in **Table 1**.

**2.1.1 Ethical approval and study registration.** This study protocol was approved by the Medical Ethics Committee of the First Teaching Hospital of Tianjin University of Traditional Chinese Medicine (item number: TYLL2023[Z]004) and was registered with the China Clinical Trial Registry (item number: ChiCTR2300071386)

**2.1.2 Informed consent.** The study will be carried out following the Declaration of Helsinki formulated by the World Medical Association and relevant clinical research regulations in China. Before patients are enrolled in this study, the researcher is responsible for fully and comprehensively introducing the purpose, procedure, and possible risks of this study to him or his designated representative in written form. Patients should be made aware of their right to withdraw at any time. All participants will obtain written informed consent before enrollment. The researcher will ensure that each patient signs an informed consent form before participating in the study, which will be kept in the study file. Patients who may be pregnant must be informed that if they get pregnant during the study, the study may endanger the fetus. Before taking part in this study, the patient should agree to use contraception during the study.

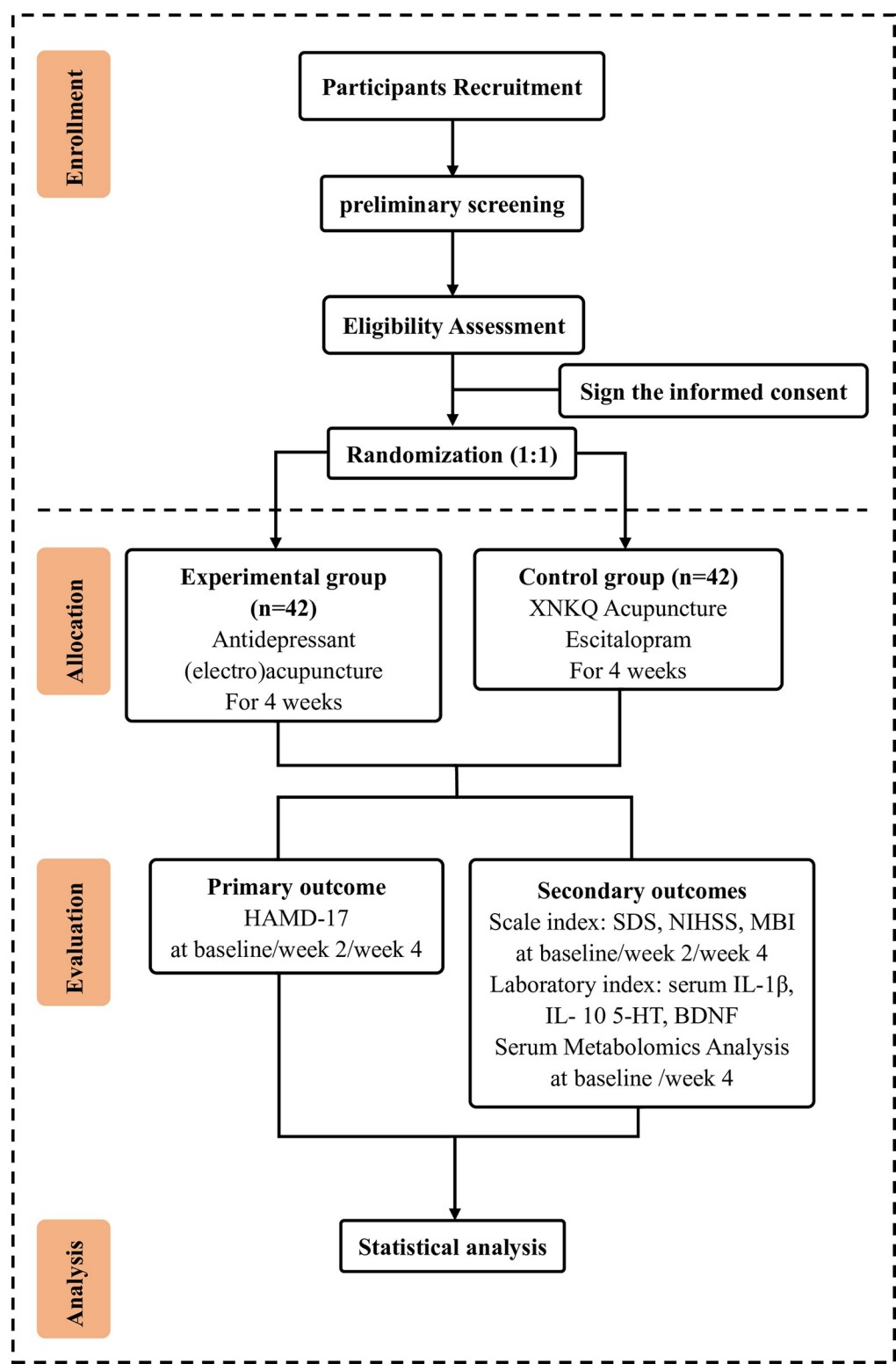

**Fig 1. The flow chart of this trial.**

**Table 1. The chart of enrollment and assessment.**

| Timepoint | Study period | | | |
|---|---|---|---|---|
| | **Enrollment** | **Baseline** | **Treatment Phase Week 2** | **Treatment Phase Week 4** |
| **Enrollment** | | | | |
| Informed consent | × | | | |
| Demographic information | × | | | |
| Basic vital signs | × | | | |
| Medical history | × | | | |
| Merger disease | × | | | |
| Eligibility screen | × | | | |
| Randomization | | × | | |
| **Interventions** | | | | |
| Experimental group | | × | × | × |
| Control group | | × | × | × |
| **Assessments** | | | | |
| HAMD-17 | | × | × | × |
| SDS | | × | × | × |
| NIHSS | | × | × | × |
| MBI | | × | × | × |
| Serum laboratory index | | × | | × |
| Serum metabolomics | | × | | × |
| **Safety** | | | | |
| Safety evaluation | | × | × | × |
| Adverse events | | × | × | × |
| Merger medication | | × | × | × |

## 2.2 Participant recruitment

A total of 84 participants will be recruited through the acupuncture outpatient and inpatient systems, as well as advertisements of the First Affiliated Hospital of Tianjin University of Chinese Medicine. The recruitment will take the form of outpatient and inpatient acupuncture promotion, recruitment advertisements on official platforms, and legal social media. According to guidelines prepared by The American Psychiatric Association and the diagnostic criteria for mental disorders compiled by The Chinese Medical Association, patients who develop symptoms within 6 months after their first stroke will be recruited.

**2.2.1 Inclusion criteria.** Participants who meet the following criteria will be included.

1. Diagnosed with stroke and meet the diagnostic criteria for "Mood disorder characterized by depression or accompanied by severe depressive episode caused by a stroke can also be characterized by mania or mixed characteristics (DSM- Ⅳ)" and "Depressive disorder due to another medical condition" in the Diagnostic and Statistical Manual of Mental Disorders, Fifth Edition (DSM-V) [22].

2. Experience a first episode of stroke onset within 6 months.

3. Age between 35 and 80 (inclusive).

4. The HAMD score of 8 to 17 (inclusive).

5. Clear consciousness, stable vital signs, cooperative physical examination.

### 2.2.2 Exclusion criteria.

1. Patients who have received antidepressant treatment within 2 weeks or are participating in other clinical trials for antidepressants.

2. Patients who have consciousness disorder or obvious cognitive impairment function (Mini-mental State Examination score <17).

3. Patients who have severe aphasia and cannot communicate.

4. History of depression prior to this stroke event.

5. Patients with severe liver and kidney dysfunction (alanine aminotransferase more than 3 times the upper limit of normal, and serum creatinine is more than 180 μmol/L).

6. Pregnant and lactating women.

7. Intolerance or refusal to receive acupuncture or electroacupuncture.

### 2.2.3 Shedding criteria.

1. Patients who have severe adverse effects will be removed from the experiment at the discretion of the researcher.

2. Patients whose condition progresses during the experiment, influencing efficacy and safety decisions.

3. Patients with low compliance (80% treatment compliance) or patients who received treatments not part of this trial.

### 2.2.4 Termination criteria.

1. Severe safety issues occurred during the trial.

2. The situation where the study does not have any clinical value, i.e., The effect of acupuncture or electroacupuncture is much lower than that of modern medicine on the market.

## 2.3 Randomization

The randomization sequence will be generated by an independent statistician with the PROC-PLAN procedure of the SAS9.3 statistical analysis system, which will not be involved in other trial procedures. To ensure allocation concealment, the generated random allocation sequence is placed in a sequentially coded, sealed opaque envelope, which will be opened in series, and the subjects will be assigned to either the observation or control group after the researcher determines the subject's eligibility. To minimize bias, the allocation sequence cards in the envelope will be force-sensitive recording paper, and the researcher's names of eligible subjects will be inscribed on the surface of the envelope before opening.

## 2.4 Blinding

Due to the characteristics and limitations of acupuncture operations, it is burdensome to blind acupuncture physicians, and we are unable to conduct a double-blind study design. Both the researchers and the operators are acquainted with the patient groups, and all patients will be treated in the acupuncture department by two licensed and experienced acupuncturists. Acupuncturists are not involved in the study design and are not allowed to communicate with

patients about acupuncture operations and acupoint selection, and patients should also avoid communication with other patients. Because of the different acupuncture intervention methods, it is impossible to blind patients. To reduce the subjective influence, the evaluators are blinded to the distribution. In data management and statistical analysis, statisticians outside this study will be invited to participate. Admittedly, they will also be blinded to group information.

## 2.5 Interventions and comparison

All patients will receive primary rehabilitation treatment during the 4-week trial. Patients with stroke and its high-risk underlying conditions (e.g., hypertension, diabetes, coronary artery disease) will be treated with secondary care and appropriate essential medication according to China's Guidelines for the Prevention and Treatment of Cerebrovascular Diseases. The physicians involved in the trial are highly qualified physicians with master's degrees in acupuncture and moxibustion from the First Teaching Hospital of Tianjin University of Traditional Chinese Medicine, who have undergone uniform and standardized training in the operation protocol and are highly experienced. For four weeks, patients will receive (electro)acupuncture in the supine or semi-supine posture once a day, six times a week. The specific operations are as follows.

**2.5.1 Control group.** Patients in the control group will receive conventional rehabilitation treatment with medication and acupuncture for 4 weeks.

1. Acupuncture treatment based on XNKQ Acupuncture for stroke dysfunction, acupoint locations are as described in **Table 2**. Acupoints include Shuigou (DU26), bilateral Neiguan (PC6), affected side Sanyinjiao (SP6), affected side Jiquan (HT1), affected side Chize (Lu5), affected side Weizhong (BL40), affected side Hegu (LI4), affected side Zusanli (ST36). The patient is placed in a supine position. After skin disinfection, the acupuncturist holds a disposable acupuncture needle (0.25mm×40mm Hwato brand, Suzhou Medical Supplies Factory Co., Ltd., Suzhou, China) for acupuncture operation. It should be noted that XNKQ Acupuncture uses a specific operation method, which is explained in Table 2. The acupuncture method is used to "De Qi" before the acupuncture is performed. Then all needles would be removed after being retained for 30 minutes. All patients will receive treatment six times each week for four weeks.

2. Oral escitalopram tablets 10mg Qd for 4 weeks.

**2.5.2 Antidepressant acupuncture group.** The patients will be placed in a semi-supine position, i.e., the bed rocker is elevated to 45˚, and after skin disinfection, the treatment proceeds as follows.

1. The exact XNKQ acupuncture as in the control group is given, and the same acupuncture method is used to "De Qi" before the antidepressant acupuncture is performed [23]. Lifting-thrusting and twirling-rotating manipulations are acupuncture procedures used to make patients feel "De Qi," such as soreness, numbness, and swelling.

2. Antidepressant acupuncture is given. The acupoint selections and ear-treating areas are Baihui (DU20), Yintang (EX-HN3), Fengfu (DU16), and auricular concha (corresponding to the auricular point Xin (CO15) and auricular point Shen (CO10)). The specific acupoint locations and operation diagram are shown in **Table 2** and Fig 2. The specific operations are as follows. The Baihui (DU20) point is inserted to a depth of 3 to 5 mm at an angle of 15˚. The Yintang (EX-HN3) point is inserted to a depth of 10 to 15 mm at an angle of 45˚

**Table 2. Specific acupuncture methods and locations of acupoints.**

| Acupoints | Location | Insert angle | Insert depth |
|---|---|---|---|
| Baihui* (DU20) | On the head, 5 cun directly above the midpoint of the anterior hairline | 15˚ | 3 to 5 mm |
| Yintang* (EX-HN3) | On the head, at the intersection of the line between the two brows and the front midline | 45˚ | 10 to 15 mm |
| Fengfu* (DU16) | On the neck, 1 cun straight above the middle of the posterior hairline, the external occipital protuberance goes straight down, and the depression between the trapezius muscles on both sides | 45˚ | 15 to 20 mm |
| Shen* (CO10) | Located in the auriculae boat, on the ear wheel, below the foot fork, on the ear wheel foot below the rear, the 10th area of the concha | 90˚ | 2 to 3 mm |
| Xin* (CO15) | Located in the depression in the middle of the concha cavity, that is, the 15th area of the concha | 90˚ | 2 to 3 mm |
| Shuigou** (DU26) | On the face, at the intersection of the upper 1/3 and middle 1/3 of the human midsulcus. In the orbicularis muscle (Sparrow-pecking acupuncture techniques until the eyeballs are wet with tears) | 45˚ | 5 to 10 mm |
| Neiguan** (PC6) | On the palm side of the forearm, between the palmaris longus tendon and the flexor carpi radialis tendon | 90˚ | 10 to 15 mm |
| Sanyinjiao** (SP6) | On the medial side of the shank, 3 cun above the medial malleolus, by the posterior of the tibia (Make the patient's limbs twitch three times) | 45˚ | 20 to 25 mm |
| Jiquan** (HT1) | At the axilla apex, where the axillary artery beats. (Make the patient's limbs twitch three times) | 45˚ | 20 to 25 mm |
| Chize** (Lu5) | In the transverse stripes of elbow, the radial depression of biceps brachii tendon (Make the patient's limbs twitch three times) | 45˚ | 20 to 25 mm |
| Weizhong** (BL40) | Located in the posterior region of the knee, at the midpoint of the popliteal transverse stripes. (Make the patient's limbs twitch three times) | 45˚ | 20 to 25 mm |
| Hegu** (LI4) | On the dorsum of the hand, between the first and second metacarpal bones, approximately in the middle of the second metacarpal bone on the radial side | 90˚ | 20 to 25 mm |
| Zusanli** (ST36) | Lateral leg, 1 cun away from the anterior tibial crest. | 45˚ | 20 to 25 mm |

* = Acupoints in Antidepressant acupuncture group. ** = Acupoints in Control group

(Needle tip towards the root of the nose). The Fengfu (DU16) point is inserted to a depth of 15 to 20 mm at an angle of 45˚ (Needle tip towards the Laryngeal node area). The auricular concha is equivalent to the position of the auricular point Xin (CO15) and auricular point Shen (CO10) of the auricular acupoint, with a depth of 2 to 3 mm and an angle of 90˚. The two electrode clamps of the electroacupuncture instrument are then clamped to the shanks of the two acupuncture needles. Baihui (DU20) and Yintang (EX-HN3), connected to a group of electro-acupuncture, auricular acupoint Xin (CO15) and acupoint Shen (CO10) connected to another group of electro-acupuncture (Hwato SDZ-IIB type electro-acupuncture instrument, Suzhou, China). The researcher set the frequency at 2/15Hz, the intensity at 2mA, and the time at 30min, and alternately acupuncture/electro-acupuncture at bilateral auricular acupoints. All patients will be treated once a day, 6 sessions per week for 4 weeks.

**2.5.3 Sample size.** In this study, HAMD-17 is used as the leading indicator for sample size calculation. Based on the results in the preliminary clinical study, i.e., observation of the Hamilton Depression Scale score reduction (ΔHAMD) results of 6.13 ± 1.96 Mean ± SD reduction in the test group and 4.57 ± 1.87 reduction in the control group, according to the sample size calculation formula.

$$\frac{2\left(Z_{\alpha/2} + Z_{\beta}\right)^2 \times \sigma^2}{\delta^2}$$

$\alpha = 0.05$, $1\text{-}\beta = 0.9$, $\sigma = 1.96$, $\delta = (6.13\text{–}4.57) = 1.56$

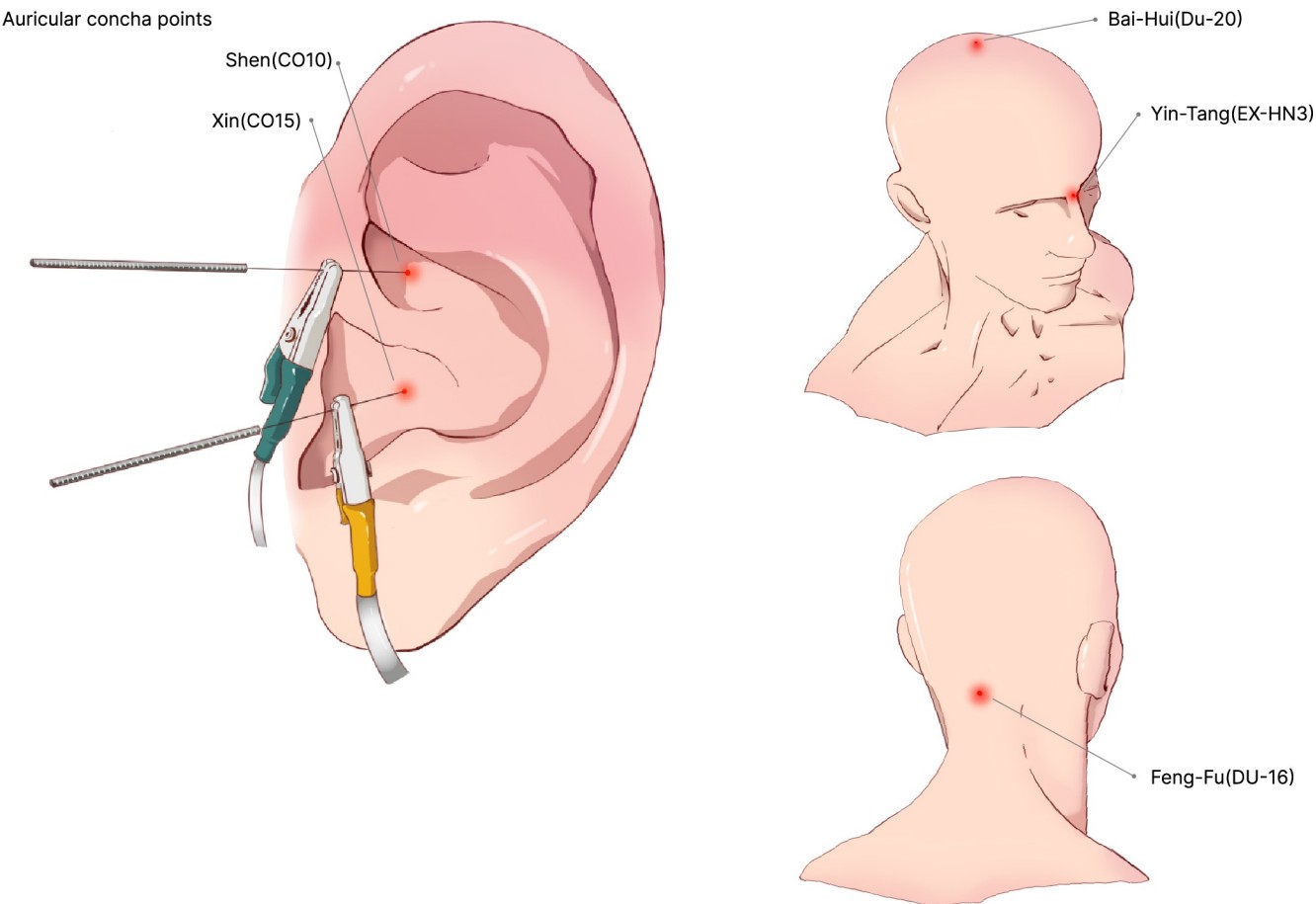

**Fig 2. Auricular concha acupoints Shen(CO10) Xin(CO15) acupoints Bai-Hui(Du20) Yin-Tang(EX-HN3) Feng-Fu(DU16).**

α is set to 0.05, the statistical power is set to 0.9, and a two-sided test is conducted. Based on the calculated dropout rate of 20%, each group will eventually require 42 patients; that is, a total of 84 participants will be randomly divided into two groups in a 1:1 ratio.

## 2.6 Outcome measures

The results cover three indicators: scale indicators, serum indicators and serum metabolomics. Scale assessments will be performed at baseline, week 2 and week 4. Serum indicators and serum metabolomics will be tested at baseline and week 4. All assessments will be performed by researchers who are blinded to treatment assignment.

**2.6.1 The primary outcome.** *2.6.1.1 Hamilton Depression Scale.* The Hamilton Depression Rating Scale (HAMD-17) is used to measure the severity of depressive symptoms in people with depressive disorders. The scoring content includes 17 symptom entries of depression, guilt, suicide, difficulty sleeping, lack of deep sleep, early awakening, work, interest, retardation, irritability, psychogenic anxiety, and physical anxiety, which is currently the most widely used.

In addition, the clinical effective rate is calculated by the reduced value of the HAMD-17 score (ΔHAMD), and the curative effect is judged. According to the Nimodipine method, the curative effect index was calculated according to the reduced score of the Hamilton Depression

scale (HAMD-17). The formula was as follows,

$$\frac{(B - A) \times 100\%}{B}$$

A = Total score after treatment. B = Total score before treatment

The curative effect is judged as follows:

Recovery: symptom score reduction $\geq$ 75%.

Significantly effective: 50%$\geq$ symptom score reduction $<$ 75%.

Effective: 25%$\geq$ symptom score reduction $<$ 50%.

Invalid: symptom score reduction $<$ 25%.

Total effective rate according to the formula

$$\frac{(R + S + E) \times 100\%}{T}$$

R = Recovery. S = Significantly effective. E = Effective. T = Total cases

**2.6.2 The second outcome.** *2.6.2.1 Self-rating Depression Scale*. Self-rating depression scale (SDS) is used to assess patients with post-stroke depression. When evaluating the curative effect, the self-evaluator should evaluate once at the baseline and once at the end of the study at two weeks and the fourth week of treatment. The SDS has 20 items, ten of which are positive and ten others negative. Before the self-evaluator evaluates, the researcher must make the subject understand the filling method of the full scale and the meaning of each question, and then make an independent self-assessment.

*2.6.2.2 National Institute of Health Stroke Scale*. The National Institute of Health stroke scale (NIHSS) [24] is used to evaluate the severity of neurological deficit symptoms in stroke patients. Researchers need to check and record the results quickly. Researchers cannot train or tell the patient hints unless necessary (such as repeatedly asking the patient to make a particular effort). If some items are not evaluated, they should be explained in detail in the table. Unassessed items can be reviewed and studied by monitoring video and other technical means and discussed with inspectors.

*2.6.2.3 Modified Barthel Index*. The modified Barthel Index (MBI) is used to evaluate the self-care ability of stroke patients; the higher the score, the stronger the patient's self-care ability, including defecation, grooming (personal hygiene), toileting, eating, transferring, walking on level ground, dressing, going up and down stairs, and bathing. Each item has five levels and associated points. The lower the score or level of an item, the more the patient's reliance on it.

*2.6.2.4 Serum laboratory index and Serum metabolomics analysis*. Serum laboratory index: determination of IL-1β, IL-10, 5-HT, and BDNF. Serum metabolomics analysis: After thawing the serum samples at room temperature, 100μL was absorbed and transferred to a fresh test tube, followed by 300μL methanol and 10μL internal standard (2- chlorophenyl alanine 2.5g/L). After the samples were mixed and centrifuged, 200μL of supernatant was sucked into the sample tube for detection by liquid chromatography-tandem mass spectrometry. The detection equipment was the AcquityTM PLC-Q-TOF-MS platform (Alfa Chemistry, USA). The advanced data features of the obtained data are extracted, processed by XCMS in R, normalized, and formed into a two-dimensional data matrix.

## 2.7 Safety evaluation and adverse events

Although acupuncture is a relatively safe treatment method with a low risk of adverse events, we require that researchers observe and ask patients about adverse reactions every time they

are treated. If there is persistent soreness and difficulty in withdrawing the needle, the treatment can be stopped at any time at the patient's discretion.

**2.7.1 Adverse event record.** In the case report form, an "adverse event record form" is set up, requiring researchers to truthfully fill in the occurrence time, severity, duration, measures taken, and outcome of adverse events.

**2.7.2 Adverse event report.** Severe adverse events are reported, and in that case, the researcher must take immediate measures to protect the safety of the subjects and report it to the unit in charge of the research and the ethics committee in time. The researcher should sign and date the report. The applicant will ensure that all reporting procedures required by laws and regulations are met.

## 2.8 Data collection and management

Before the start of the study, all researchers and acupuncturists will attend a three-day training course, including evaluation and collection, organized by the First Teaching Hospital of Tianjin University of Traditional Chinese Medicine. If participants can't complete the study, these data will also be included in the statistics and kept. The case report form (CRF) will be used in this study to collect data. All personnel involved in data collection need to be confirmed and describe their work. Clinical researchers must fill in CRF accurately, timely, completely, and standardly according to the original information. The modification of CRF data must follow standard operating procedures, leaving traces of transformation. Once errors or discrepancies are found, researchers should be informed to ensure that all data records and reports are correct and complete. If necessary, the data management plan can be updated and revised in time according to the change in the research scheme, but corresponding procedures are needed to complete it. Data managers should thoroughly check the primary and secondary effectiveness indicators and key safety indicators specified in the system to ensure the correctness and integrity of these data. Any changes in the research plan will be notified to the ethics commission and the registration center promptly. This process will be carried out under the supervision of project supervision experts.

## 2.9 Statistical analysis

Third-party statisticians who are not engaged in the allocation process or trial implementation will be invited to conduct statistical analysis and participate in the whole process, including trial design, implementation, and data analysis. Outcome measures at baseline and two-time points after the intervention will be analyzed to assess treatment effectiveness. Data will be analyzed using SPSS26.0 software according to the intention-to-treat principle of data analysis. Subgroup analysis will be performed if data for different clinical features are available. Chi-square tests will be used to assess sample characteristics and gender. Continuous variables will be summarized as mean ± standard deviation (SD). Continuous variables will be compared by the Student t-test or Wilcoxon rank sum test, and categorical variables will be reached by the Pearson chi-square test or Wilcoxon rank sum test reported by percentage. Data conforming to the normal distribution will be analyzed by one-way analysis of variance. The comparison of each group will be analyzed by T-test. A non-parametric rank sum test and the Mann-Whitney U test will be used to assess measurement data that does not have a normal distribution or homogeneous variance. A bilateral 5% significance level will be used, and the corresponding 95% confidence interval will be calculated as far as possible. P-value $<0.05$ would be set as the significance level, and the difference would be considered statistically significant.

**2.9.1 Quality control.** Qualified acupuncturists, doctors, researchers, and statisticians in the First Teaching Hospital of Tianjin University of Traditional Chinese Medicine monitored

and revised this experimental scheme. They also execute process quality control and data quality control, including compliance control implemented by the researcher, subject compliance control, and follow-up information integrity control, as well as creating a quality control team. Data quality control includes the composition of data supervisors, responsibilities and workflow, input process, interim analysis plan, research termination standard, data verification, data cleaning, etc. In addition, there will be regular supervision by clinical experts. Upon completion of the study, we will provide repository information for our data, all data underlying the findings will be made available as part of the manuscript or supporting information, or deposited in a public repository where the data will be recorded.

## 3 Discussion

Although several animal tests on PSD in existing studies focus on the pathogenesis of PSD, there is limited clinical evidence [25]. It is challenging to diagnose and assess PSD prior to clinical treatment. According to studies, only 5% of stroke survivors are diagnosed and treated for depression in routine clinical practice, as they frequently suffer cognitive and communicative impairments [26–28]. The difficulty of treatment is increased especially because of late diagnosis and screening, resulting in a relatively high proportion of small sample sizes and low-quality research in most clinical studies. The general lack of clinical evidence in the international field is also a massive problem in the future [21, 29].

Multiple biological and social psychological factors impact the pathogenesis of PSD, and its therapeutic mechanism has been intensively studied. There are studies discussing promoting neuroplasticity [30] and the perspective of central or peripheral immune inflammation [31]. Recent studies [3, 32] demonstrated that pro-inflammatory and anti-inflammatory mediators and the Sirt1/ NF-κB pathway are all implicated in the M1/M2 polarization of microglia in ischemic brain tissue. We previously discovered that "NRBP1", "SIRT1", "BDNF", "MAPK3", "CREB1" and key biological pathways such as NF-B, PI3K/AKT activation, and MAPK are critical for the incidence and development of PSD biomarkers [20]. Among them, SIRT1 can reduce the binding of NF-κB to inflammatory nuclear genes through the deacetylation of RelA/P65, a subunit of NF-κB, thereby reducing the production of inflammatory cytokines such as TNF-α and IL-1β. In the following animal experiments, we hypothesized that acupuncture might improve the inflammatory environment of PSD and regulate the polarization direction of M1/M2 microglia, and found experimentally and clinically that the expression levels of plasma SIRT1 (P value < = 0.05) and P65 (P value < = 0.05) were significantly lower after acupuncture treatment than in disease controls. Acupuncture treatment down-regulated the Sirt1/NF-κB pathway to inhibit the inflammatory responses, initially confirmed the intervention of acupuncture on these key pathways and targets, and to a certain extent, the mechanism by which acupuncture action promotes anti-inflammatory factor levels and neuronal protective regeneration, and thus improves PSD symptoms.

In this investigation, the antidepressants used for the control group are selective 5-hydroxytryptamine reuptake inhibitors (SSRIs), which have been shown to lower the risk of depression but may also raise the risk of fractures, seizures, brain hemorrhage, and gastrointestinal symptoms [9, 33]. The acupoints selected for the experimental group in this study are particular acupuncture points (Baihui(GV20), Yintang(EX-HN3), Fengfu(GV16), auricular concha, i.e., ear acupuncture acupoints Xin(CO15) and acupoints Shen(CO10)) in addition to the XNKQ acupuncture method of electro-acupuncture connection. Recent studies by Cai W et al [13] have indicated that electro-acupuncture (EA) appears to be an effective and safe therapy for PSD, with EA reducing depressive symptoms and improving neurological function. Several RCTs have proven the clinical efficacy of (electro)acupuncture in Baihui (GV20) and Yintang

(EX-HN3) [34–38]. This is in close relation to the activation of the prefrontal lobes of the brain, which has been proven. In particular, Zhang, Z, J et al demonstrated that the use of an integrative acupuncture approach (electroacupuncture stimulation with body acupuncture) for the treatment of PSD, especially electrostimulation, reduced PSD and cognitive dysfunction [39, 40]. Fengfu (GV16) acupoint is located near the high cervical spine on the back and improves the blood supply of the vertebral artery. Several animal research and clinical data in recent years have demonstrated the effect of percutaneous vagal nerve stimulation on depression and even severe depression for the selection of auricular acupoints [3, 41, 42]. Among them, Wang, J. Y et al used the auricular concha region for stimulation and found an α7nAchR/NF-κB pathway associated with its antidepressant effect. These several pieces of high-quality evidence illustrate the importance of treatment of the auricular region for depression.

Both non-invasive and invasive auricular therapy have been shown to be effective and safe in two RCTs [12, 38]. In a meta-analysis, auricular acupuncture was also considered one of the safe treatments for non-drug interventions to improve PSD symptoms [29]. In the current study, many Meta-analyses of acupuncture combined with drugs for PSD suggest the potential advantages of acupuncture combined with drugs [43–45]. In an umbrella review, acupuncture and electroacupuncture plus conventional treatment with escitalopram, citalopram, sertraline, and fluoxetine showed no severe adverse events in patients with PSD [46]. Adverse reactions and side effects are likely to be mild, and several systematic reviews suggest that acupuncture and electroacupuncture treatments are either somewhat more effective than antidepressants or not significantly different from them, with fewer adverse events and side effects and a higher safety profile [47–49]. At the same time, it is also mentioned that low-quality studies account for a significant proportion, and that multicenter and large-sample double- or multi-blind randomized controlled clinical trials are still urgently needed.

In this trial, while assessing the clinical efficacy of acupuncture and electroacupuncture in the treatment of PSD, the effect of acupuncture on inflammatory factors in patients with PSD will be observed, and the mechanism of action will be further explored in depth to provide a theoretical and clinical basis for clinical acupuncture treatment of PSD. This study will observe multiple indicators as secondary outcomes that may have implications for enriching the theoretical and clinical evaluation of PSD diagnosis and treatment, leading to a more effective and multilevel exploration of the close association between acupuncture for PSD and mechanisms such as anti-inflammatory effects.

There are several limitations to this study. It is a single-center study, and the regional restriction of patients brings bias into the experimental research. Additionally, due to the limit of the trial date period, long-term patient follow-up is not achievable. However, we will remain committed to stepping up our efforts to standardize the treatment of PSD patients, to work to reduce negative effects and biases in the research process, and to provide high-quality evidence for future research in the acupuncture treatment of the PSD patient population as part of an integrative and complementary alternative medicine program for the treatment of PSD in the medical field.

## Supporting information

**S1 Checklist. CONSORT 2010 checklist of information to include when reporting a randomised trial\*.**
(DOC)

**S1 Protocol.**
(PDF)

 

## Author Contributions

**Conceptualization:** Dawei Ran, Yu Wang.

**Funding acquisition:** Menghan Li.

**Investigation:** Qian Zhu, Yuzheng Du.

**Methodology:** Lei Shi.

**Software:** Xiaoli Song.

**Writing – review & editing:** Xinming Yang.

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
