## [Decision Letter · Decision Letter 0]

12 Dec 2023

PONE-D-23-31081Efficacy of Acupuncture in the Treatment of Post-stroke Depression: A Study Protocol of a Randomized Controlled Trial

PLOS ONE

Dear Dr. YANG,

Thank you for submitting your manuscript to PLOS ONE. After careful consideration, we feel that it has merit but does not fully meet PLOS ONE’s publication criteria as it currently stands. Therefore, we invite you to submit a revised version of the manuscript that addresses the points raised during the review process.

We look forward to receiving your revised manuscript.

Kind regards,

Yung-Hsiang Chen, Ph.D.

Academic Editor

PLOS ONE

Journal Requirements:

"Tianjin Education Commission Research Program Project (Grant No. 2021KJ150)"

**Additional Editor Comments:**

Thank you for submitting the following manuscript to PLOS ONE.

Please revise the manuscript according to the reviewers' comments and upload the revised file.

Reviewers' comments:

Reviewer's Responses to Questions

**Comments to the Author**

1. Does the manuscript provide a valid rationale for the proposed study, with clearly identified and justified research questions?

Reviewer #1: Yes

Reviewer #2: Yes

2. Is the protocol technically sound and planned in a manner that will lead to a meaningful outcome and allow testing the stated hypotheses?

Reviewer #1: Yes

Reviewer #2: Yes

3. Is the methodology feasible and described in sufficient detail to allow the work to be replicable?

Reviewer #1: Yes

Reviewer #2: Yes

4. Have the authors described where all data underlying the findings will be made available when the study is complete?

Reviewer #1: Yes

Reviewer #2: No

5. Is the manuscript presented in an intelligible fashion and written in standard English?

Reviewer #1: Yes

Reviewer #2: Yes

6. Review Comments to the Author

You may also provide optional suggestions and comments to authors that they might find helpful in planning their study.

Reviewer #1: There are minor English errors throughout the proposed protocol.

Introduction:

Stroke increases the risk of PSD, which is also a negative factor in stroke recovery

Since PSD is defined as “post stroke depression” stroke does not increase the risk of PSD, it is a required factor to have PSD. This needs to be re-written.

Methods: tense should be changed to consistently be future tense:

For example: One group received XNKQ acupuncture mainly, and escitalopram was taken… This should be re-written as One group WILL receive XNKQ acupuncture mainly, and escitalopram will be taken….

This tense issue remains throughout this section.

Adverse event report

Suppose severe adverse events appear in the experiment. In that case, the researcher must take

immediate measures to protect the safety…

This needs to be written without the “suppose” and adverse events happen to subjects not “appear in the experiment.” Please re-write this section.

Reviewer #2: Dear editors

The following is my opinion, and the purpose of this trial is to demonstrate the efficacy of acupuncture in PSD. Patients six months after stroke were selected for this study. Depression is one of the most important sequelae of stroke, and PSD often appears within 3 months of stroke. The golden rehabilitation period for stroke is 3-6 months after stroke, and post-stroke depression will reduce the patient's willingness to do rehabilitation, slow down the recovery from stroke, and even miss the opportunity of the golden rehabilitation period for stroke. Acupuncture for depression is a proven treatment that can be recommended. The study was not completed, and there were no experimental results in the text. The number of people who actually participated in and dropped out of the trial, adverse reactions, etc. Ask questions about the design of the study are as follows.

1 Will the patients in the study lose the best opportunity for treatment after six months of stroke?

2 Inclusion criteria 4 on page 5, Select subjects The HAMD score of 8 to 17 with mild depression. Will the HAMD score be less sensitive to changes in the severity of depression in mild cases, and will it affect the outcome analysis?

3 Depending on your design, Acupuncturists and subjects cannot communicate with each other. If a patient has adverse effects after needling, such as persistent soreness or difficulty withdrawing needles, will the patient withdraw from the trial, or will there be other treatments?

4 Are the acupuncture points fixed or can they be adjusted, and will the subjects receive acupuncture and electroacupuncture at the same time?

5 Isn't the medication necessarily the same for different subjects?

6 Table 2 shows the acupuncture points of both the experimental group and the control group, please color distinguish them in Table 2 to avoid confusion.

7 According to the text, the acupuncture of the control group did not administer "De Qi".

8 XingnaoKaiqiao (XNKQ) acupuncture is usually used in patients who are unconscious in the acute phase of stroke, where techniques such as Sparrow-pecking acupuncture techniques in Shuigou (DU26) on the upper lip can cause extreme pain and tears and are subjects with depression in remission of stroke too irritated.

7. PLOS authors have the option to publish the peer review history of their article (what does this mean?). If published, this will include your full peer review and any attached files.

Reviewer #1: **Yes: **Jennifer Brett

Reviewer #2: No

---

## [Author Response · Author response to Decision Letter 0]

14 Feb 2024

Dear reviewers,

 We deeply appreciate your review work. 

Thank you for raising constructive questions about this study, which can make it more scientific. We have also deeply reflected on and made revisions. The following are responses to the questions raised by two reviewers. If you have any questions, please criticize and contact us.

Answers to Reviewer 1's Questions:

Q: There are minor English errors throughout the proposed protocol.

Introduction:

Stroke increases the risk of PSD, which is also a negative factor in stroke recovery

Since PSD is defined as “post stroke depression” stroke does not increase the risk of PSD, it is a required factor to have PSD. This needs to be re-written.

Methods: tense should be changed to consistently be future tense:

For example: One group received XNKQ acupuncture mainly, and escitalopram was taken… This should be re-written as One group WILL receive XNKQ acupuncture mainly, and escitalopram will be taken….

This tense issue remains throughout this section.

Adverse event report

Suppose severe adverse events appear in the experiment. In that case, the researcher must take

immediate measures to protect the safety…

This needs to be written without the “suppose” and adverse events happen to subjects not “appear in the experiment.” Please re-write this section.

Re: Thank you for your careful guidance. We regret the presentation errors, grammatical and tense errors in the manuscript, and have revised and proofread the manuscript in its entirety, including deleting the relevant content in the introduction section regarding the relationship between stroke and PSD. The tense in the Methods section has been changed to future tense. The Adverse Event Report section has been rewritten.

Answers to Reviewer 2's Questions:

1. Will the patients in the study lose the best opportunity for treatment after six months of stroke?

Re: Thank you for your concern in the long-term treatment of stroke in the patients who will be involved in this study. The timing of stroke rehabilitation is critical, and there are statistics that 8 to 12 weeks after stroke is a period of rapid recovery from physical dysfunction in patients. If early rehabilitation can be carried out as early as six months after the onset of the disease, the survival rate of patients will increase to over 90% [1]. It can be confirmed that there are still opportunities for treatment even after six months. Taking into account the actual clinical efficacy and the patient's own needs for treatment, after a six-month stroke prognosis and rehabilitation, in addition to the basic medication that will continue to be taken under the supervision of the physician, the treatment will continue to have acupuncture outpatient treatment by the responsible physician. It is certain that this study will not delay the clinical treatment available to any subjects.

[1] Shao Limin. Evaluation of the effect of medical and nursing joint health education intervention on nursing extension service for patients discharged from hospital with cerebral infarction[J]. Modern practical medicine,2018,30(07):968-969.

2. Inclusion criteria 4 on page 5, Select subjects The HAMD score of 8 to 17 with mild depression. Will the HAMD score be less sensitive to changes in the severity of depression in mild cases, and will it affect the outcome analysis?

Re: With regard to the choice of depression scales, we believe that the HAMD is necessary, the HAMD scale is the "gold standard" for the diagnosis and assessment of depression, and the scores of the 17-item HAMD were derived from the ratings according to the conversion method described by Endicott [1], and in the conclusions of the study by Zimmerman [2] et al. patients with mild depression had significantly lower HAMD scores than those with moderate depression, and those with moderate depression had significantly lower HAMD scores than those with major depression. The HAMD threshold that maximized the sum of sensitivity and specificity was 17 in the comparison of mild and moderate depression, and 24 in the comparison of moderate and severe depression, thus suggesting that HAMD is sensitive in mild depression. In addition, Helmreich [3] et al. compared the Inventory of Depressive Symptomatology(IDS-C) and HAMD scales concerning sensitivity to changes in antidepressant treatment, and although mild depression was more sensitive on the PSDIDS-C subscale, the subscales did not cover all aspects of depression and the benefit ratios, and the IDS-C scales themselves are less frequently used in diagnostic assessment of PSD, so we still selected the classical scale HAMD.

[1] Endicott J, Cohen J, Nee J, Fleiss J, Sarantakos S. Hamilton Depression Rating Scale: Extracted From Regular and Change Versions of the Schedule for Affective Disorders and Schizophrenia. Arch Gen Psychiatry. 1981;38(1):98–103. doi:10.1001/archpsyc.1981.01780260100011

[2] Zimmerman M, Martinez JH, Young D, Chelminski I, Dalrymple K. Severity classification on the Hamilton Depression Rating Scale. J Affect Disord. 2013;150(2):384-388. doi:10.1016/j.jad.2013.04.028

[3] Helmreich, I., Wagner, S., Mergl, R. et al. Sensitivity to changes during antidepressant treatment: a comparison of unidimensional subscales of the Inventory of Depressive Symptomatology (IDS-C) and the Hamilton Depression Rating Scale (HAMD) in patients with mild major, minor or subsyndromal depression. Eur Arch Psychiatry Clin Neurosci 262, 291–304 (2012). https://doi.org/10.1007/s00406-011-0263-x

3. Depending on your design, Acupuncturists and subjects cannot communicate with each other. If a patient has adverse effects after needling, such as persistent soreness or difficulty withdrawing needles, will the patient withdraw from the trial, or will there be other treatments?

Re: The scope of the inability to communicate in the design of this manuscript refers to the inability to communicate between the subjects and the acupuncturists about the plan, and the subject's inability to communicate with other subjects, for which we have rewritten in the revised manuscript due to inappropriate language, rewritten in the 2.4 Blinding section. If subjects experience adverse reactions after acupuncture, such as persistent soreness or difficulty in removing the needle, we can refer to the additional content on the treatment of adverse reactions in section 2.7 Safety Evaluation and Adverse Events

4. Are the acupuncture points fixed or can they be adjusted, and will the subjects receive acupuncture and electroacupuncture at the same time?

Re: Once the study is underway, all acupuncture points will be fixed, including those in the antidepressant acupuncture group and those in the control group, while subjects in the antidepressant acupuncture group will receive both "XNKQ" acupuncture and electroacupuncture.

5. Isn't the medication necessarily the same for different subjects?

Re: In Section 2.5 Interventions and Comparison, we set out the need for physicians responsible for treating high-risk underlying conditions for stroke to provide basic symptomatic treatment in response to the patient's symptoms.Basic therapeutic drugs are not necessarily the same for different subjects, however, the principle of medication is consistent across subjects, for example, all will be taking antiplatelet drugs and lipid-lowering drugs, and there will also be for hypertension, diabetes, etc. There may also be a superimposition of different types of underlying diseases. In several studies [1-3] on the use of acupuncture in the treatment of post-stroke-related disorders, both control and experimental groups were given conventional treatments, including symptomatic treatments such as infection prevention, vasodilatation, and anticoagulation, which suggests that receiving different symptomatic primary treatments does not have a significant impact on the existing treatment protocols or on the subjects themselves.

[1] Chu Jiaxin,Peng Fei, Li Jiaojiao et al. Clinical effect of waking up the brain and opening up the mind acupuncture combined with repetitive transcranial magnetic stimulation in the treatment of post-stroke thalamic pain[J]. Chinese Medicine Herald,2023,20(32): 88-92.DOI:10.20047/j.issn1673-7210.2023.32.19.

[2] Yang Daohai,Miao Yongjuan,Xu Shasha et al. Effects of waking up the brain and opening up the mind acupuncture method on the morphology of oropharynx and sleep breathing parameters in patients with obstructive sleep apnoea syndrome after stroke[J]. Shanghai Journal of Acupuncture and Moxibustion,2023,42(06):565-569.DOI:10.13460/j.issn.1005-0957.2023.06.0565.

[3] Yu Zhuoting,Chen Yangfu,Xu MiMi. Analysis of therapeutic effect of waking up the brain and opening up the mind in treating patients with hemiplegia after stroke by acupuncture[J]. Liaoning Journal of Chinese Medicine,2023,50(09):192-195.DOI:10.13192/j.issn.1000-1719.2023.09.051.

6. Table 2 shows the acupuncture points of both the experimental group and the control group, please color distinguish them in Table 2 to avoid confusion.

Re: The lack of clarity in the grouping of acupuncture points in Table 2 is an omission, and to show the grouping of the points, we have added a description of the grouping of acupoints in Table 2, *=Acupoints in the Antidepressant acupuncture group. **=Acupoints in the Control group

7. According to the text, the acupuncture of the control group did not administer "De Qi".

Re: Thank you for your careful monitoring of the requirement of "De Qi" as an important step in the practice of acupuncture, we have added the following sentence to section 2.5.1 of the manuscript “The acupuncture method is used to “De Qi” before the acupuncture is performed” , and to add an explanation of "De Qi" in section 2.5.2.

8. XingnaoKaiqiao (XNKQ) acupuncture is usually used in patients who are unconscious in the acute phase of stroke, where techniques such as Sparrow-pecking acupuncture techniques in Shuigou (DU26) on the upper lip can cause extreme pain and tears and are subjects with depression in remission of stroke too irritated.

Re: Shuigou(DU 26)is from "Zhen Jiu Jia Yi Jing", belonging to the DuMai, in the face of the human middle groove of the upper 1/3 and the middle 1/3 point of the intersection. In a study by Zhang [1], specific acupuncture points of the "XNKQ" acupuncture method combined with medication for the treatment of PSD improved the patients' depression and ability to perform daily life and reduced the corresponding inflammatory response. Huang [2] et al. also found the effect of acupuncture at Shuigou on the function of limbs and intracranial haemodynamics of patients recovering from a stroke, in their study, when patients recovering from stroke were treated with acupuncture at Shuigou acupoint and Baihui combined with acupuncture at the foot transporting sensory area, the levels of NT-3 and NGF increased, indicating that the treatment of patients recovering from stroke with acupuncture at Shuigou acupoint and Baihui combined with acupuncture at the foot transporting sensory area was able to improve the patients' neurological function. intracranial haemodynamics. Yan [3] found the Shuigou point to be a high-frequency point in a study on the matching pattern of acupoints selected for acupuncture treatment of post-stroke depression. In addition, in a recent study by Li [4] et al, high-level evidence was provided that the "XNKQ" acupuncture method is effective in the treatment of post-stroke motor aphasia, which is a long-term prognosis.

[1] Zhang Chuanwen, Min Xirui. Effects of waking up the brain and opening up the orifices acupuncture combined with wuling capsule in treating post-stroke depression on autophagy-related protein levels[J]. Shanghai Journal of Acupuncture and Moxibustion,2023,42(02):121-126.DOI:10.13460/j.issn.1005-0957.2023.02.0121.

[2] Huang Shiwei,LI Gui Ying,Yang Liang. Effects of acupuncture Shuigou and Baihui points combined with foot transporting sensory zone on limb function and intracranial haemodynamics in patients recovering from stroke[J]. New Chinese Medicine,2022,54(21):178-182.DOI:10.13457/j.cnki.jncm.2022.21.040.

[3] Yan KH,Ren RX,Fan WT et al. Analysis of the pattern of acupuncture point selection and matching in the treatment of post-stroke depression by acupuncture[J]. Journal of Integrated Cardiovascular and Cerebrovascular Diseases of Chinese and Western Medicine,2023,21(03):421-425.

[4] Li B, Deng S, Zhuo B, et al. Effect of Acupuncture vs Sham Acupuncture on Patients With Poststroke Motor Aphasia: A Randomized Clinical Trial. JAMA Netw Open. 2024;7(1):e2352580. Published 2024 Jan 2. doi:10.1001/jamanetworkopen.2023.52580

---

## [Decision Letter · Decision Letter 1]

28 Feb 2024

PONE-D-23-31081R1Efficacy of Acupuncture in the Treatment of Post-stroke Depression: A Study Protocol of a Randomized Controlled TrialPLOS ONE

Dear Dr. YANG,

Thank you for submitting your manuscript to PLOS ONE. After careful consideration, we feel that it has merit but does not fully meet PLOS ONE’s publication criteria as it currently stands. Therefore, we invite you to submit a revised version of the manuscript that addresses the points raised during the review process.

Thank you for submitting the following manuscript to PLOS ONE.

Please revise the manuscript according to the reviewers' comments and upload the revised file.

We look forward to receiving your revised manuscript.

Kind regards,

Yung-Hsiang Chen, Ph.D.

Academic Editor

PLOS ONE

Journal Requirements:

Additional Editor Comments:

Thank you for submitting the following manuscript to PLOS ONE.

Please revise the manuscript according to the reviewers' comments and upload the revised file.

Reviewers' comments:

Reviewer's Responses to Questions

**Comments to the Author**

1. Does the manuscript provide a valid rationale for the proposed study, with clearly identified and justified research questions?

Reviewer #1: Yes

Reviewer #2: Yes

2. Is the protocol technically sound and planned in a manner that will lead to a meaningful outcome and allow testing the stated hypotheses?

Reviewer #1: Yes

Reviewer #2: Yes

3. Is the methodology feasible and described in sufficient detail to allow the work to be replicable?

Reviewer #1: Yes

Reviewer #2: Yes

4. Have the authors described where all data underlying the findings will be made available when the study is complete?

Reviewer #1: Yes

Reviewer #2: Yes

5. Is the manuscript presented in an intelligible fashion and written in standard English?

Reviewer #1: Yes

Reviewer #2: Yes

6. Review Comments to the Author

You may also provide optional suggestions and comments to authors that they might find helpful in planning their study.

Reviewer #1: Acupoint identification and naming issues remain in sections 2.5.1 and 2.5.2

In 2.5.1: Shuigou (DU26) is said to be needled bilaterally. DU26 has only one location on the midline. Are two needles to be placed?

In 2.5.2 use of the acupoint names is inconsistent with Table 2. In the table the Du Mai points are labeled Du 16 and Du 20. In 2.5.2, acupoint names use “GV” rather than DU: “Baihui (GV20)… Fengfu (GV16)”

Please change all to “DU” in 2.5.2.

Reviewer #2: The author answered the previous question and can accept its correction. Question 2 asked earlier Inclusion criteria 4 Select subjects The HAMD score of 8 to 17 with mild depression. It may affect the difference, resulting in little difference between the two groups, but it will also need to be verified by clinical trials.

7. PLOS authors have the option to publish the peer review history of their article (what does this mean?). If published, this will include your full peer review and any attached files.

Reviewer #1: No

Reviewer #2: No

---

## [Author Response · Author response to Decision Letter 1]

8 Apr 2024

Dear reviewers,

 We deeply appreciate your review work. 

Thank you for raising constructive questions about this study. We have also deeply reflected on and made revisions. The following are responses to the questions raised by two reviewers. If you have any questions, please criticize and contact us.

Answers to Reviewer 1's Questions:

Q: Acupoint identification and naming issues remain in sections 2.5.1 and 2.5.2. In 2.5.1: Shuigou (DU26) is said to be needled bilaterally. DU26 has only one location on the midline. Are two needles to be placed? In 2.5.2 use of the acupoint names is inconsistent with Table 2. In the table the Du Mai points are labeled Du 16 and Du 20. In 2.5.2, acupoint names use “GV” rather than DU: “Baihui (GV20)… Fengfu (GV16)” Please change all to “DU” in 2.5.2.

Re: Thank you for your careful guidance. We apologize for these errors and have now standardized the descriptions of acupoints in 2.5.1, 2.5.2 and Table 2, with Baihui being standardized as DU20, Yintang as EX-HN3 and Fengfu as DU16. The standardization of the terminology and names of the locations is very important and will be taken care of in future studies. We will be careful in future studies. 

Answers to Reviewer 2's Questions:

Q: The author answered the previous question and can accept its correction. Question 2 asked earlier Inclusion criteria 4 Select subjects The HAMD score of 8 to 17 with mild depression. It may affect the difference, resulting in little difference between the two groups, but it will also need to be verified by clinical trials. 

Re: Thank you very much for raising this thought-provoking question. In fact, when designing the experiment, we mainly wanted to observe the effect of acupuncture therapy on mild depression after stroke. If we include patients with scores above 17 and severe depression, medication intervention is necessary, and the existing research protocol does not comply with clinical guidelines and ethical requirements. However, the comparison with drug efficacy is not the problem that our research aims to solve. We aim to explore the significance of acupuncture intervention in patients with mild depression after stroke based on the anti-inflammatory mechanism of acupuncture. The comparison of drug efficacy may be the plan for the next step of research. Of course, the question you raised is also a reality, so we have read a lot of related research. We found that a published study also used the same inclusion criteria for HAMD segments[1]. And they also obtained statistically significant results. Moreover, sample size is also a factor that affects the statistical significance of the results. This study used 35 patients in each group, while our study sample consisted of 42 patients in each group. So we believe that this setting is reasonable. If, as you mentioned, the research results do not show significant differences, we will carefully analyze the reasons and discuss them, which also provides an opportunity for future research. 

[1] Sun Q, Xu H, Zhang W, Zhou Y, Lv Y. Behavioral Activation Therapy for Subthreshold Depression in Stroke Patients: An Exploratory Randomized Controlled Trial. Neuropsychiatr Dis Treat. 2022 Nov 29;18:2795-2805. doi: 10.2147/NDT.S392403.

---

## [Decision Letter · Decision Letter 2]

22 Apr 2024

Efficacy of Acupuncture in the Treatment of Post-stroke Depression: A Study Protocol of a Randomized Controlled Trial

PONE-D-23-31081R2

Dear Dr. Yang,

We’re pleased to inform you that your manuscript has been judged scientifically suitable for publication and will be formally accepted for publication once it meets all outstanding technical requirements.

Kind regards,

Yung-Hsiang Chen, Ph.D.

Academic Editor

PLOS ONE

Additional Editor Comments (optional):

Congratulations on the acceptance of your manuscript, and thank you for your interest in submitting your work to PLOS ONE.

Reviewers' comments:

Reviewer's Responses to Questions

**Comments to the Author**

1. Does the manuscript provide a valid rationale for the proposed study, with clearly identified and justified research questions?

Reviewer #1: Yes

2. Is the protocol technically sound and planned in a manner that will lead to a meaningful outcome and allow testing the stated hypotheses?

Reviewer #1: Yes

3. Is the methodology feasible and described in sufficient detail to allow the work to be replicable?

Reviewer #1: Yes

4. Have the authors described where all data underlying the findings will be made available when the study is complete?

Reviewer #1: Yes

5. Is the manuscript presented in an intelligible fashion and written in standard English?

Reviewer #1: Yes

6. Review Comments to the Author

You may also provide optional suggestions and comments to authors that they might find helpful in planning their study.

Reviewer #1: The updated submission has addressed all concerns previously identified by this reviewer.

No additional corrections needed.

7. PLOS authors have the option to publish the peer review history of their article (what does this mean?). If published, this will include your full peer review and any attached files.

Reviewer #1: **Yes: **Jennifer Brett

---

## [Editor Report · Acceptance letter]

3 May 2024

PONE-D-23-31081R2 

PLOS ONE

Dear Dr. Yang, 

I'm pleased to inform you that your manuscript has been deemed suitable for publication in PLOS ONE. Congratulations! Your manuscript is now being handed over to our production team.

Kind regards, 

on behalf of

Dr. Yung-Hsiang Chen 

Academic Editor

PLOS ONE